# Spark Plasma Sintering Behavior of Nb-Mo-Si Alloy Powders Fabricated by Hydrogenation-Dehydrogenation Method

**DOI:** 10.3390/ma12213549

**Published:** 2019-10-29

**Authors:** Sung Yong Lee, Ki Beom Park, Jang-Won Kang, Yanghoo Kim, Hyun-Su Kang, Tae Kwon Ha, Seok-Hong Min, Hyung-Ki Park

**Affiliations:** 1Gangwon Regional Division, Korea Institute of Industrial Technology, Gangneung 25440, Korea; sylee494@kitech.re.kr (S.Y.L.); hope92430@kitech.re.kr (K.B.P.); jwk1313@kitech.re.kr (J.-W.K.); kyhoo05@kitech.re.kr (Y.K.); kanghsoo@kitech.re.kr (H.-S.K.); 2Department of Advanced Metal and Materials Engineering, Gangneung-Wonju National University, Gangneung 25457, Korea; Taekwonh@gwnu.ac.kr

**Keywords:** Nb-Mo-Si alloy, Nb silicide-based composite, powder, spark plasma sintering, sintering behavior

## Abstract

In this study, the sintering behaviors of Nb-6Mo-20Si-3Cr (at percentage) in situ composite powders were studied. The Nb alloy powder was fabricated by a hydrogenation-dehydrogenation method, and both the alloy ingot and powders consisted of two phases: An Nb metal phase and the α-Nb_5_Si_3_ phase. Consolidation of the alloy powders was performed at 1500, 1600, and 1700 °C using spark plasma sintering, and the microstructures and phases formed at various sintering temperatures were analyzed. Micropores were observed in the compact sintered at 1500 °C due to the lack of complete densification at that temperature. The densification was completed at 1600 °C and the microstructure was slightly coarsened at 1700 °C compared to the microstructure of the compact sintered at 1600 °C. The microstructures prepared by the powder metallurgy method were finer than the microstructure of the ingot prepared by the casting method. The phase formation behavior varied according to the sintering temperature. Specifically, the α-Nb_5_Si_3_ phase, which is a stable structure of the Nb_5_Si_3_ phase at a low temperature, was transformed to the β-Nb_5_Si_3_ phase (which is stable at a high temperature) with an increasing sintering temperature.

## 1. Introduction

The gas turbine efficiency improves when increasing the turbine inlet temperature. Accordingly, many studies have been carried out to increase the available working temperatures of high temperature alloys. Along these lines, many researchers have studied ways to develop high temperature alloys, and past studies mainly focused on Ni-based superalloys. However, the development of refractory metals based on Nb has been carried out in recent years in response to the demand for next-generation high-temperature alloys that can be used above the usable limit temperature of Ni-based superalloys [1,2,3].

Nb alloy consists of an Nb metal phase and Nb silicide phases formed by the addition of Si [4,5]. Therefore, the Nb alloy is referred to as Nb silicide-based in situ composite. The formation of Nb silicide phases in the alloy both improves the high-temperature oxidation resistance and enhances the creep resistance at high temperatures [6,7,8]. Si in the Nb silicide is oxidized to form SiO_2_ when Nb alloys are exposed to the high-temperature oxidation atmosphere [9]. Then, Si in the silicide diffuses along the surface to form a SiO_2_ film over the whole surface region, which increases oxidation resistance. In addition, the silicide phases have good high temperature mechanical properties such as strength and creep resistance, which increases the available working temperature of the Nb alloy [10].

In recent years, studies have focused on replacing some of the Nb with Mo to improve the available working temperature of the Nb alloy. Wang et al. [11] studied the microstructure and high-temperature mechanical properties of Nb-Mo-Si in situ composites. They reported that the compressive strengths of Nb-Mo-Si alloys reached about 1000 MPa at a temperature of 1523 K.

Despite these advantages, Nb alloys have many difficulties associated with processing and component fabrication. First, since the melting temperature of Nb is 2750 K, it is difficult to manufacture parts through casting technology. In addition, it is hard to process part of the shape by cutting because of the low fracture toughness and poor machinability of the silicide in the Nb alloy [12,13]. Therefore, it is necessary to study the manufacturing technology of its components using powder metallurgy in the development of Nb alloys. However, there are few studies on the powder production and sintering behavior of Nb alloys.

There are various methods for powder fabrication of metals, including a hydrogenation-dehydrogenation (HDH) method for mass production of powder at a low cost. The HDH is a method of annealing a metal in a hydrogen atmosphere to form a metal hydride, wherein the metal hydride is pulverized into powders using its brittle nature. Representatively, Ti is an element that is phase-transformed to TiH_2_ during annealing in a hydrogen atmosphere. Therefore, its powders could be prepared using the HDH method [14]. This method is thermodynamically applicable to metals capable of reacting with hydrogen to form metal hydrides. Among refractory metals, some studies reported that powders were fabricated by adopting the HDH method in pure Ta [15] and pure Nb [16].

Based on this background, we conducted a study on the powder production and sintering behavior of Nb-Mo-Si silicide-based composite. The alloy powders have a composition of Nb-6Mo-20Si-3Cr and were prepared by the HDH method. In addition, they were consolidated by a spark plasma sintering process. We investigated the formation of microstructures and phases in the compact with respect to the sintering temperature.

## 2. Materials and Methods

An Nb-6Mo-20Si-3Cr (at percentage) alloy ingot was prepared by a vacuum arc re-melting method. The alloy melted in a cold copper crucible and it was re-melted 5 times. Afterward, the cast ingot was heat treated at 1200 °C for 10 h in a high vacuum atmosphere to homogenize the chemical composition. The chemical composition of the ingot was examined by wavelength-dispersive X-ray fluorescence spectrometry, and the result is shown in Table 1. Oxygen concentration of the cast ingot was examined using an oxygen analyzer (LECO, 736 series, LECO, San Jose, MI, USA) and the oxygen concentration of the ingot after homogenization annealing was measured to 0.001 wt percentage.

The Nb-Mo-Si alloy powders were prepared using an HDH process. The ingot was heat treated in a 100% hydrogen atmosphere for hydrogenation. After that, the hydrogenated ingot was pulverized by jaw crushing and ball milling, and the powders were sieved to a size ranging from 37 to 90 μm. The powders were heat-treated in a vacuum atmosphere for dehydrogenation. The detailed process for powder fabrication is provided in the literature [17].

The powders fabricated by the HDH method were consolidated by a spark plasma sintering process. The powders were placed into a graphite mold with a diameter of 10 mm and a height of 12 mm. After that, they were densified for 10 min at temperatures ranging from 1500 to 1700 °C under a pressure of 40 MPa. The consolidation process was performed in a high vacuum atmosphere, and the heating rate was kept constant at 200 °C/min.

The microstructures of the ingot, the powders, and the consolidated compacts were investigated using a field emission-scanning electron microscope (FE-SEM) (QUANTA FEG 250, FEI, Hillsboro, OR, USA) equipped with an energy-dispersive spectrometer (EDS) (Octane Elite EDS, EDAX, Mahwah, NJ, USA). Phase analyses of the samples were carried out by an X-ray diffractometer (XRD) with Cu Kα radiation. The distribution of the powder size was analyzed using a powder size analyzer (Mastersizer 3000, Malvern Panalytical, UK). To compare mechanical property of the samples, the hardness of the ingot and the compacts sintered at 1500, 1600, and 1700 °C were analyzed using a Rockwell hardness tester. An average value was obtained by testing each sample five times.

## 3. Results and Discussion

Figure 1a shows the microstructures of the initial alloy ingot. Left and right images are the low and high magnification microstructures, respectively. Based on the microstructure, the Nb alloy ingot consisted of two phases, and this result is consistent with a previous report [11]. To examine the composition of the phase, the microstructure of the right side in Figure 1a was investigated by energy dispersive X-ray spectroscopy (EDS) area mapping. Figure 1b shows the EDS mapping results. In the bright region, Mo and Nb concentrations were high, while the dark region was rich in Si. The bright and dark regions are, thus, identified as the Nb metal and silicide phases, respectively. 

To identify the crystal structure, the ingot was examined by X-ray diffractometer (XRD), and the result is presented in Figure 2. As shown in the XRD result, the crystal structure of the ingot was revealed to be the Nb metal and the α-Nb_5_Si_3_ silicide phase. From the XRD analysis, it can be inferred that the Nb metal phase has 3.263 Å of the lattice parameter. Table 2 shows the percentages of the two phases in the ingot, which were determined from the XRD data using the Rietveld method. The compositions of the Nb metal phase and the α-Nb_5_Si_3_ silicide phase were 51.9% and 48.1%, respectively. After hydrogen annealing, the Nb solid solution phase was hydrogenated and its peak positions were changed. However, in the case of the Nb_5_Si_3_ phase, the positions of the peaks were maintained. After removal of hydrogen through dehydrogenation annealing, the XRD pattern was returned to the same position as the initial ingot.

Figure 3a shows the morphologies of the initial powders for the spark plasma sintering. The powders had an irregular morphology, which is often observed in Ti powders fabricated by the HDH method [18]. In a previous study, Lee et al. reported the fabrication of the Nb-Mo-Si alloy powders using the HDH reaction [17]. Since Nb is a strong metal hydride-forming element from a thermodynamic perspective, hydrogen can enter into Nb metal during annealing in the hydrogen atmosphere, which results in the Nb hydride phase [19,20]. After hydrogen annealing, the Nb alloy was easily pulverized to a powder due to the brittle nature of the Nb hydride. Then, the conversion from the hydride to Nb-Mo-Si alloy powders could be achieved by removing hydrogen through a vacuum heat treatment. Figure 3b represents the powder size distribution of the powders in Figure 3a, where the particle sizes in terms of d_10_, d_50_, and d_90_ were measured to be 36.5, 58.6, and 88.9 μm, respectively.

XRD analysis was performed to investigate the phases of the powder, and the result is shown in Figure 2. Like the ingot, the powders were composed of two phases: An Nb solid solution phase and an Nb_5_Si_3_ silicide phase. An analysis of the percentages of phases in the ingot and powders showed that the percentage of the Nb solid solution and the α-Nb_5_Si_3_ silicide phase were the same. However, the peak intensities were different in the XRD data for the ingot and powders. This is due to the texture effect, i.e., the formation of a coarse microstructure in the ingot [13]. From the XRD analysis, it can be inferred that the Nb metal phase has 3.263 Å of the lattice parameter, whose value is the same with that of the initial ingot.

Figure 3c is the cross-sectional image of the powder. Many cracks were observed inside the powder, mainly in the Nb_5_Si_3_ phases, which is the dark region. During hydrogenation, the Nb phase absorbs hydrogen and undergoes volume expansion [21]. Based on the XRD results (Figure 2), the lattice parameter of Nb phase in the ingot was 3.263 Å and it was increased to 3.372 Å after hydrogen annealing. Thus, a volume of about 10% was expanded by hydrogen absorption after the hydrogenation. At the same time, the Nb_5_Si_3_ phase breaks and cracks due to its low fracture toughness [12].

The powders were consolidated using spark plasma sintering to investigate the sintering behavior of the Nb alloy powders. Figure 4 shows the microstructures of the spark plasma sintered compacts. Figure 4a–c are the microstructures of compacts sintered at temperatures of 1500, 1600, and 1700 °C, respectively, with left and right images showing low and high magnifications.

Figure 4a is the microstructure of the compact sintered at 1500 °C. Micropores appeared as black dots in the microstructure, which means that the temperature of 1500 °C was not high enough for the powders to be completely densified. The microstructure was refined when compared with the microstructure of the ingot in Figure 1. The initial ingot was pulverized after hydrogenation to fabricate the powders. A refined microstructure was produced in the powder metallurgy process since the coarse grains in the ingot were crushed into small sizes and they did not grow sufficiently at a temperature of 1500 °C.

Figure 4b shows the microstructure of the compact sintered at 1600 °C. The sizes of the microstructure at a sintering temperature of 1600 °C were similar to that sintered at 1500 °C. However, the shape of the interface between the Nb metal and the Nb silicide phase became smoother. No micropores were observed in Figure 4b, which means that the powders were fully consolidated at 1600 °C.

The microstructure of the Nb alloys affects the oxidation resistance at high temperatures [22]. Si was added to the Nb alloy to enhance the high-temperature oxidation and creep resistance, which results in the formation of silicide phases [23]. At high temperatures, a SiO_2_ film is formed on the surface, which increases the oxidation resistance by acting as a protective layer [10]. In an oxidizing atmosphere, the SiO_2_ film is formed first on the silicide phase, and Si is diffused into the Nb solid solution phase along the surface. This diffusion covers the whole surface of the alloy. The distance between the silicide phases decreased as the microstructure was refined [24]. Therefore, increased oxidation resistance could be achieved by adopting the powder metallurgy process since the Si diffusion distance required to cover the Nb solid solution phase could be reduced by microstructure refinement. 

Figure 4c represents the microstructure of the compact sintered at a temperature of 1700 °C. The microstructure was coarsened compared to that of the compacts sintered at 1500 and 1600 °C due to the increased sintering temperature. The microstructures of the sintered compacts (Figure 4) show that the sintering at 1600 °C is appropriate for the Nb-Mo-Si-Cr alloy because a fully consolidated and refined microstructure could be achieved at that temperature.

Figure 5 shows the EDS area analysis results of the 1600 °C sintered compact. The area shown in Figure 5 is the same as the microstructure of the right side of Figure 4b. Like the data for the ingot (Figure 1), the bright region had a low concentration of Si. The Si concentration of the dark region was much higher than that of the bright region. These concentration mapping results indicate that the bright and dark regions are the Nb metal and Nb silicide phases, respectively.

In order to confirm the change in the mechanical property due to such a microstructural change, the hardness of the ingot and the compacts sintered at 1500, 1600, and 1700 °C were analyzed. The hardness of the initial ingot was 113.2 HRB and the hardness of the compacts sintered at 1500, 1600, and 1700 °C was 124.6, 125.2, and 124.6 HRB, respectively. In the case of the sintered samples, the hardness increased compared to the initial ingot, but the hardness was similar among the sintered samples. Since the microstructure of the sintered samples is much finer than that of the ingot, the hardness increased.

To investigate the change in phases and their composition with respect to the sintering temperature, the compacts sintered at 1500, 1600, and 1700 °C were analyzed by XRD, and the results are shown in Figure 6a. The XRD peak position and intensity changed with the sintering temperature, which means that the phase formation in the compacts was affected by the temperature. From the XRD results, the sintered compacts mainly consisted of the Nb metal phase (ICDD # 98-064-5060) and the Nb_5_Si_3_ phases. However, there are three Nb_5_Si_3_ phases with different crystal structures of two tetragonal (α-Nb_5_Si_3_, β-Nb_5_Si_3_) structures and one hexagonal structure. Both the α-Nb_5_Si_3_ and β-Nb_5_Si_3_ phases have a tetragonal crystal structure. However, lattice parameters a, b, and c of α-Nb_5_Si_3_ are 6.56, 6.56, and 11.86 Å, respectively (international centre for diffraction data (ICDD) #98-001-6774), while those of β-Nb_5_Si_3_ are 10.02, 10.02, and 5.07 Å, respectively (ICDD #98-064-5411).

To precisely examine the change in the crystal structure, the patterns in the 2θ ranged from 35° to 45° and were expanded, as shown in Figure 6b. From the XRD result of the ignition ingot (Figure 2), the ingot and the powders were composed of the Nb metal and the α-Nb_5_Si_3_ (tetragonal structure) phase. However, the compact sintered at 1500 °C was composed of three phases: the two main phases of the Nb metal and the α-Nb_5_Si_3_ (tetragonal structure) and one minor phase of the Nb_5_Si_3_ (hexagonal structure). The α-Nb_5_Si_3_ phase transformed to the Nb_5_Si_3_ phase, which has a hexagonal structure during the spark plasma sintering process, because the Nb_5_Si_3_ (hexagonal structure) was not present in the initial powder (Figure 2).

The β-Nb_5_Si_3_ phase was newly observed in the compact sintered at 1600 °C, and the peak intensity of the α-Nb_5_Si_3_ phase slightly decreased. In the compact sintered at 1700 °C, the peak intensity of the β-Nb_5_Si_3_ phase drastically increased, and the peak of the α-Nb_5_Si_3_ phase disappeared. 

Two Nb_5_Si_3_ phases with tetragonal structures were presented based on the Nb-Si binary phase diagram [1], and the α-Nb_5_Si_3_ phase (which is stable at low temperature) was transformed to the β-Nb_5_Si_3_ phase at the high temperature. Therefore, the β-Nb_5_Si_3_ phase appeared and increased in terms of the phase percentage as the sintering temperature increased. At 1700 °C, the α-Nb_5_Si_3_ phase eventually fully transformed to the β-Nb_5_Si_3_ phase.

The percentages of each phase in the ingot and the compacts sintered at 1500 °C, 1600 °C, and 1700 °C were obtained by the Rietveld method based on the XRD data, and the results are presented in Table 2. The weighted profile R-factors (R_wp_) in the Rietveld analysis ranged from 15% to 19%. The phase percentages of Table 2 were represented by wt %. Comparing the phases of the four samples, the percentage of the Nb solid solution was between 50–51% in all samples. In the compact sintered at 1500 °C, the α-Nb_5_Si_3_ phase was transformed, forming 3.2% of the Nb_5_Si_3_ phase with the hexagonal structure. When the sintering temperature was 1600 °C, the percentage of the Nb_5_Si_3_ with the hexagonal structure increased to 7.2%, and the β-Nb_5_Si_3_ phase was present at 3.3%. In the 1700 °C sintered sample, the fraction of the Nb_5_Si_3_ phase with the hexagonal structure remained similar to that at 1600 °C, but the α-Nb_5_Si_3_ phase disappeared. In addition, 42.0% of the β-Nb_5_Si_3_ phase was observed. These results indicate that the Nb solid solution phase did not change with the sintering temperature, but the overall phase ratio changed as the silicide phase changed with temperature.

## 4. Conclusions

In this study, Nb-6Mo-20Si-3Cr in situ composite powders were prepared, and their consolidation behaviors were analyzed with respect to temperature during the spark plasma sintering process.
(1)To investigate the sintering behavior of the alloy powders, powders were prepared using the hydrogenation-dehydration reaction. Irregularly-shaped powders with a d_50_ value of 58.6 μm were fabricated using this method.(2)Micropores were observed in the compact sintered at 1500 °C because the densification was not fully completed. At 1600 °C, the densification was completed, and the microstructure of the compact was finer than that of the ingot. At the sintering temperature of 1700 °C, the microstructure was slightly coarsened compared to the microstructure of the compacts sintered at 1500 °C and 1600 °C.(3)The Nb_5_Si_3_ (hexagonal structure) phase was present in the compact prepared by the powder metallurgy method, which was different from the initial ingot and powders. In addition, the α-Nb_5_Si_3_ phase transformed to the β-Nb_5_Si_3_ phase as temperature increased. The β-Nb_5_Si_3_ phase began to form at a sintering temperature of 1600 °C. At a sintering temperature of 1700 °C, all the α-Nb_5_Si_3_ phase disappeared, and the β-Nb_5_Si_3_ phase formed as the main silicide phase.

## Figures and Tables

**Figure 1 materials-12-03549-f001:**
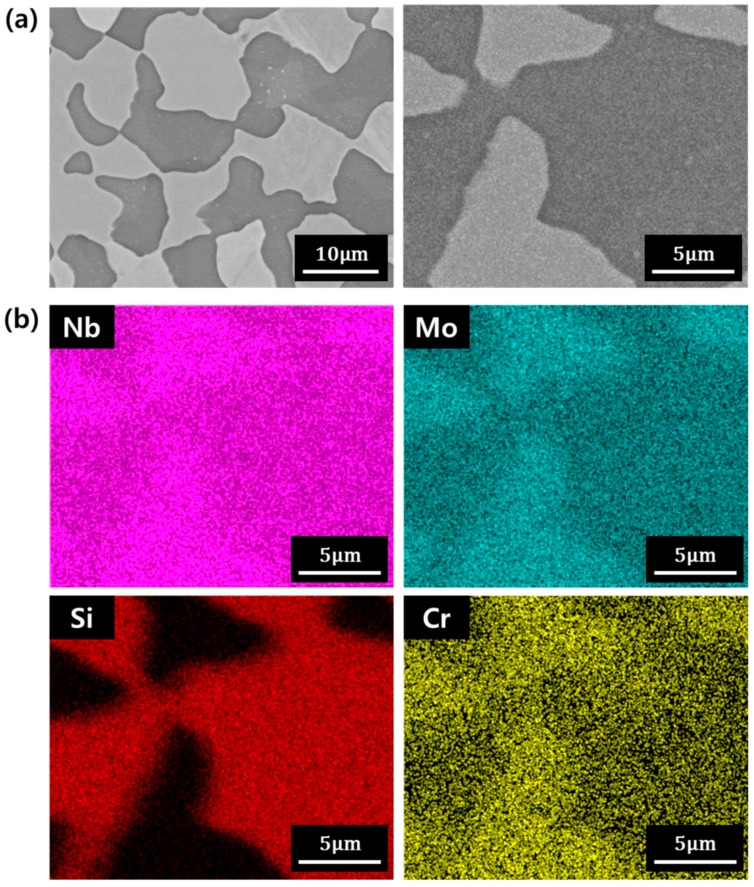
(**a**) Microstructure of the ingot observed via backscattered electron imaging. (**b**) EDS mapping results of the microstructure on the right side in Figure 1a.

**Figure 2 materials-12-03549-f002:**
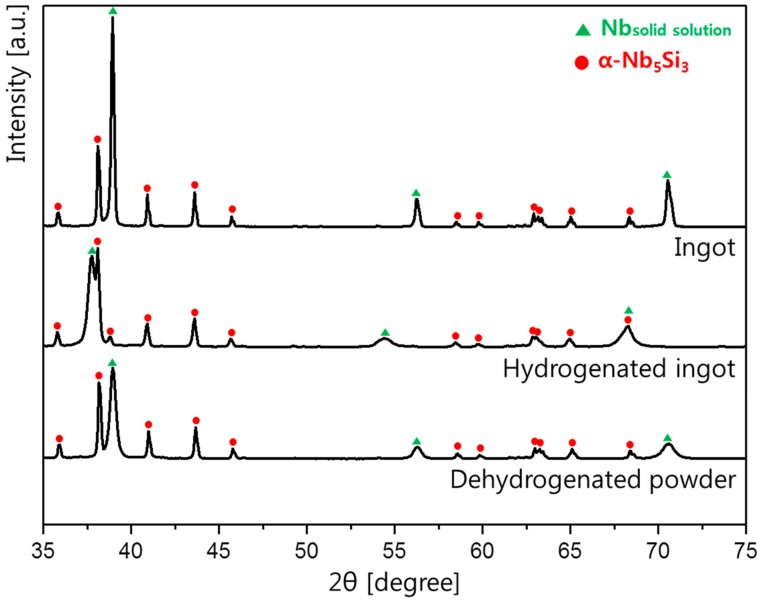
XRD patterns showing crystal structures of samples: the ingot, hydrogenated ingot, and dehydrogenated powders.

**Figure 3 materials-12-03549-f003:**
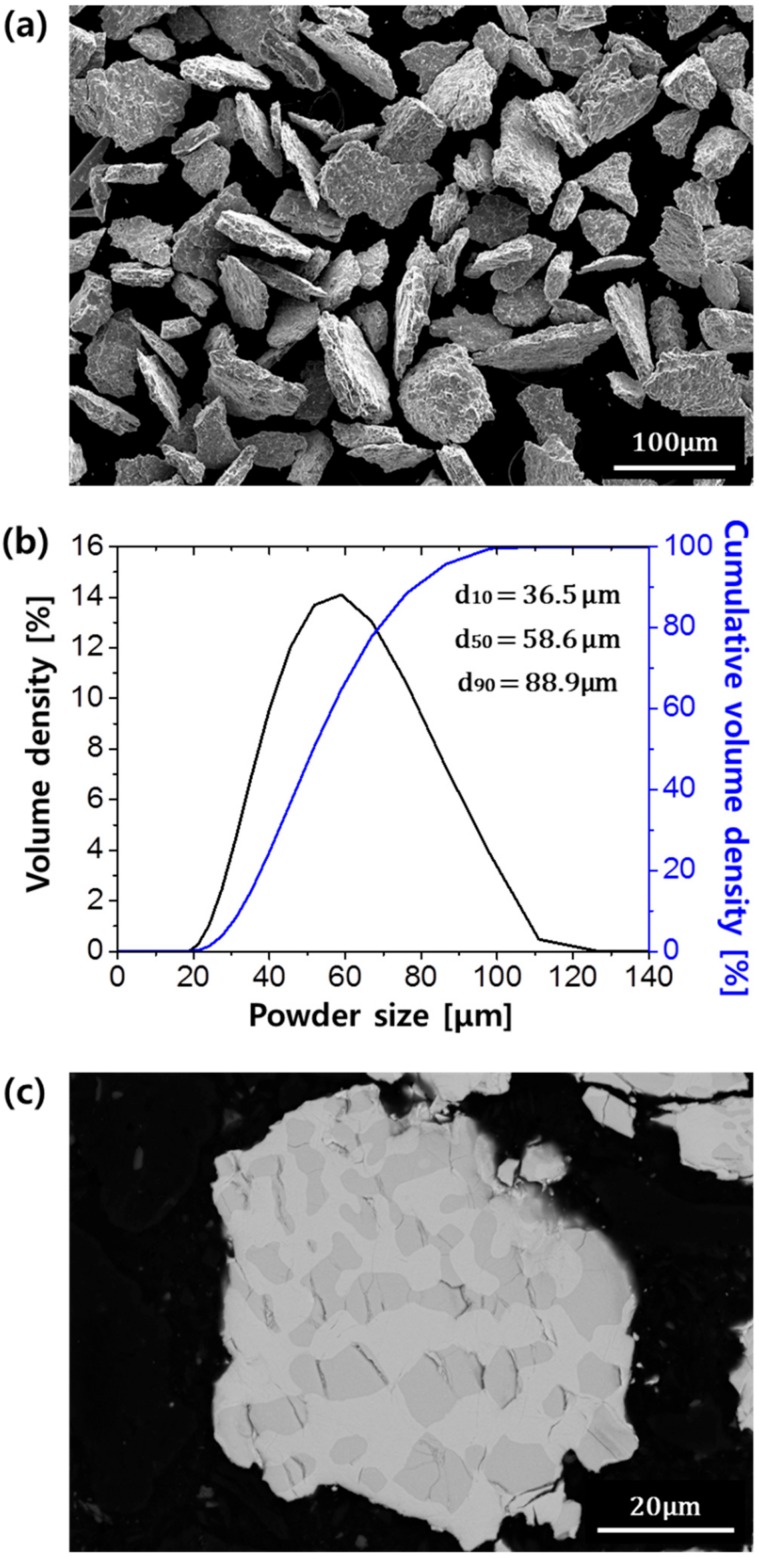
(**a**) The scanning electron microscope (SEM) image shows the morphology of the dehydrogenated powders and (**b**) their size distribution, and (**c**) the cross-sectional image shows the inner structure of the powder.

**Figure 4 materials-12-03549-f004:**
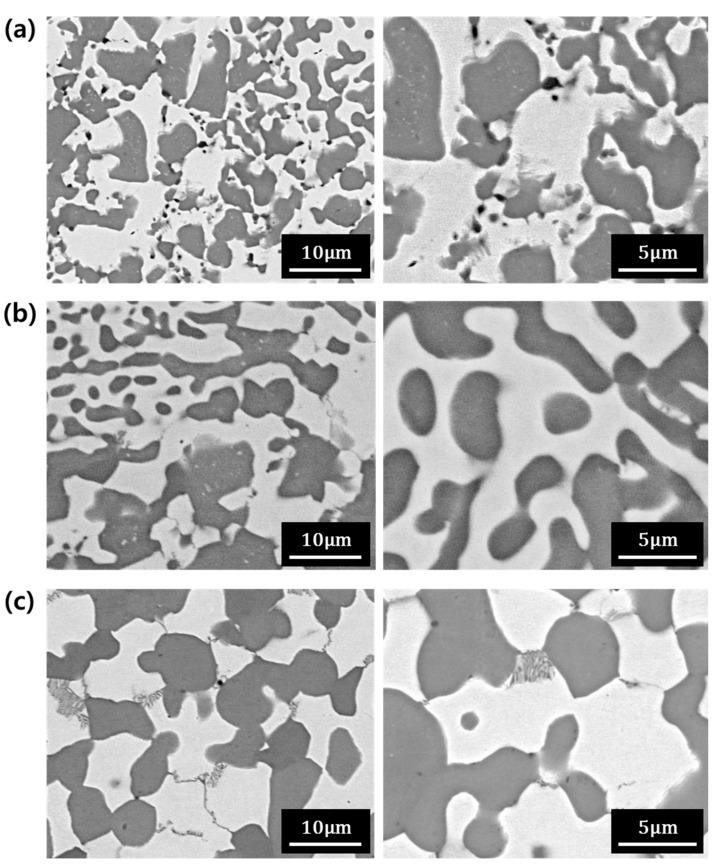
(**a**) Microstructure of the compacts sintered at (**a**) 1500, (**b**) 1600, and (**c**) 1700 °C as observed via backscattered electron imaging.

**Figure 5 materials-12-03549-f005:**
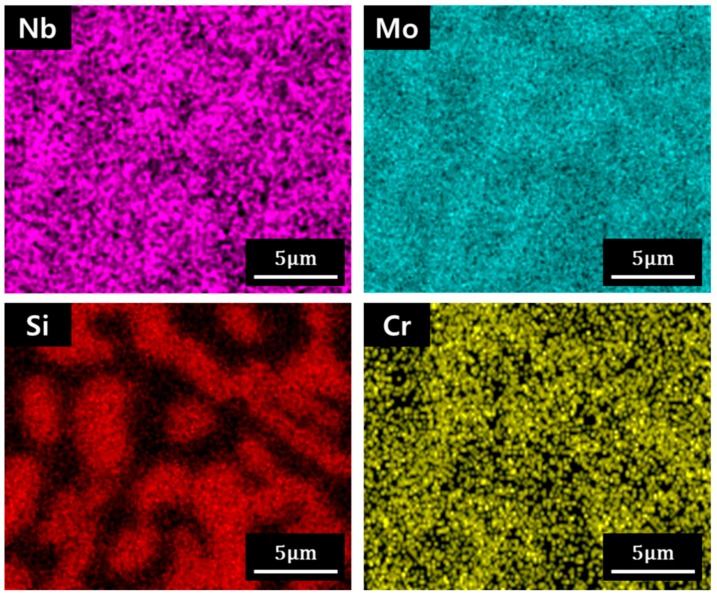
EDS mapping result of the compact sintered at 1600 °C. Mapping was performed in the same region as the microstructure image on the right side in Figure 4b.

**Figure 6 materials-12-03549-f006:**
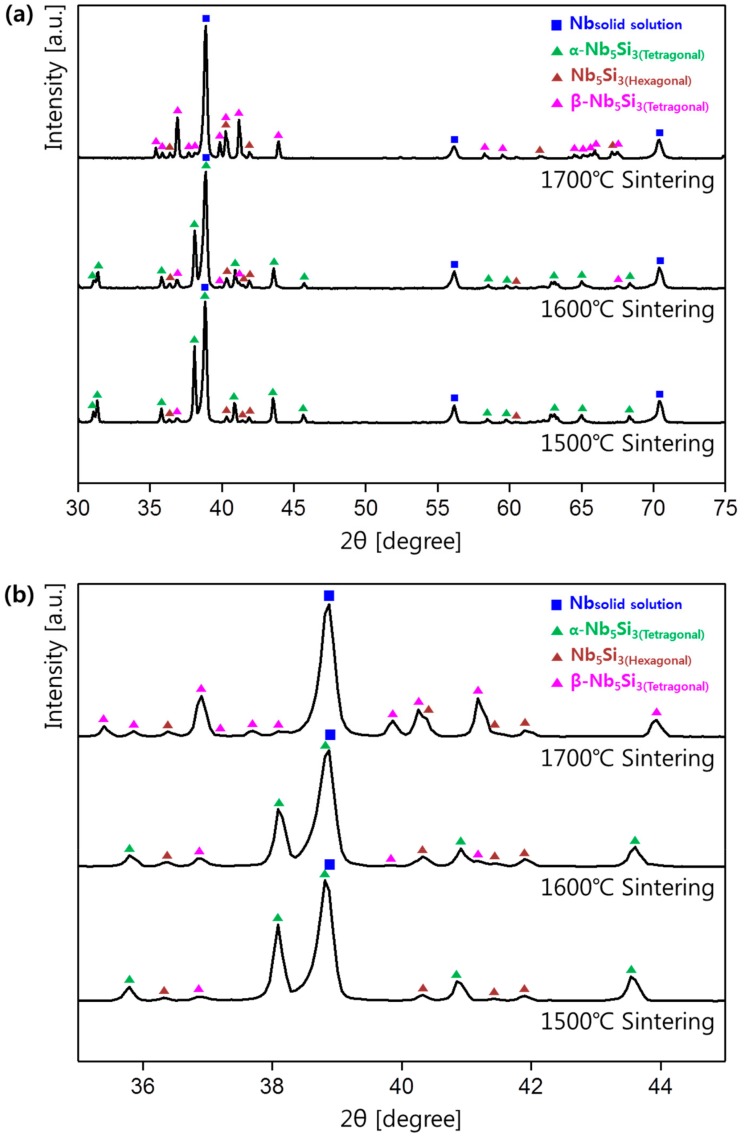
(**a**) XRD patterns of the compacts sintered at 1500°, 1600°, and 1700° and (**b**) expanded XRD patterns in the 2θ range of 35–45°.

**Table 1 materials-12-03549-t001:** Chemical composition of the initial alloy ingot (at percentage).

Element	Nb	Mo	Si	Cr
Atomic%	71.2	6.1	19.8	2.9

**Table 2 materials-12-03549-t002:** The percentages of phases in the ingot and the compacts sintered at 1500 °C, 1600 °C, and 1700 °C were analyzed using the Rietveld method (wt %).

Sample	Nb Solid Solution	α-Nb_5_Si_3_ (Tetragonal)	Nb_5_Si_3_ (Hexagonal)	β-Nb_5_Si_3_ (Tetragonal)
Ingot	51.9	48.1	-	-
Compact sintered at 1500 °C	51.0	46.8	3.2	-
Compact sintered at 1600 °C	50.2	39.3	7.2	3.3
Compact sintered at 1700 °C	50.6	-	7.4	42.0

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
