# Peer review of "Spark Plasma Sintering Behavior of Nb-Mo-Si Alloy Powders Fabricated by Hydrogenation-Dehydrogenation Method"

_materials, 2019, doi:10.3390/ma12213549_

Round 1
Reviewer 1 Report
Hydrogenation-dehydrogenation approach to processing refractory alloys seems to have nice potential for further applications. The authors described the methods well but need to improve their manuscript for further consideration, as follows:
(1) Need to describe prior art on similar processing method if it exists. This information can be searched.
(2) The Nb alloys are designed for high temperature applications. Did the authors test the robustness of the microstructure when subject to a prolonged annealing.
(3) There is no information on the mechanical properties.
Author Response
Replies to the comments of reviewer #1
Comment: Need to describe prior art on similar processing method if it exists. This information can be searched.
Answer:
- As mentioned by the reviewer, we added new references and following sentences in the introduction part.
Ref. [14] : C.R.F. Azevedo; D. Rodrigues; F.B. Neto. Ti–Al–V powder metallurgy (PM) via the hydrogenation–dehydrogenation (HDH) process, JALCOM 2003, 353, 217-227.
Ref. [15] : K.T. Park; J.H. Park; J.H. Yoon; J.E. Lee; I.K. Park. Temperature-dependent Ta hydride formation for recycling of Ta scraps: Experimental and thermodynamic investigations, Inter. J. Refrac. Metals & Hard Mater. 2017, 65, 83-87.
Ref. [16] : H.R.Z. Sandim; A.F. Padilha. On the sinterability of commercial-purity niobium, Key Eng. Mater. 2001, 189-191, 296-301.
“There are various methods for powder fabrication of metals, among which a hydrogenation-dehydrogenation (HDH) method is a technique for mass production of powder at low cost. The HDH is a method of annealing a metal in a hydrogen atmosphere to form a metal hydride, wherein the metal hydride is pulverized into powders using its brittle nature. Representatively, Ti is an element that is phase-transformed to TiH2 during annealing in a hydrogen atmosphere; therefore, its powders could be prepared using the HDH method [14]. This method is thermodynamically applicable to metals capable of reacting with hydrogen to form metal hydrides. Among refractory metals, some studies were reported that powders were fabricated by adopting the HDH method in pure Ta [15] and pure Nb [16].” (line 56)
Comment: The Nb alloys are designed for high temperature applications. Did the authors test the robustness of the microstructure when subject to a prolonged annealing?
Answer:
- This study has focused on the powder fabrication of Nb-Mo-Si based in situ composite and its sintering behavior. At present, no research has been conducted on robustness of the microstructures during long-term heat treatment. We plan to conduct research on robustness in the future, and please understand that there is no data at present.
Comment: There is no information on the mechanical properties.
Answer:
- As mentioned by the reviewer, we analyzed Rockwell hardness of ingot and compacts sintered at 1500, 1600, and 1700°C and added following sentences.
“To compare mechanical property of the samples, the hardness of the ingot and the compacts sintered at 1500, 1600, and 1700°C were analyzed using Rockwell hardness tester. An average value was obtained by testing five times in each sample.” (line 96)
“In order to confirm the change in mechanical property due to such microstructure change, the hardness of the ingot and the compacts sintered at 1500, 1600, and 1700°C were analyzed. The hardness of the initial ingot was 113.2 HRB and the hardness of the compacts sintered at 1500, 1600 and 1700 °C was 124.6, 125.2, and 124.6 HRB, respectively. In the case of the sintered samples, the hardness increased compared to the initial ingot, but the hardness was similar among the sintered samples. Since the microstructure of the sintered samples is much finer than that of the ingot, hardness was increased.” (line 196)

Reviewer 2 Report
The paper Spark plasma sintering behavior of Nb-Mo-Si based in situ composite powders fabricated by hydrogenation-dehydrogenation reaction gives a concise description of an experiment performed on NbMoSi composite. The paper is well written, the language is excellent, the text follows a main line.
Here you can find some comments to the text:
Table I - Could you add the Oxygen concentration, or the measurement error, so as we can see, that the initial O content is below that value, which cannot be measured?
line 174 - "To investigate the change in phase with respect to the sintering temperatur" the word composition should be inserted
Table 2 contains data from the sintered compacts, which are discussed later in the text. Could you please the table after Figure 6? The table shows wt% of phases, please add this information, that these are wt% also to the text (Line 206-210)
LIne 181-182- (ICDD # xx-xx-xxx) - should be deleted, The lattice parameters are in Angstroms not nanometers
You discussed at several places, that the Si phase oxidizes, why there are no maps for O content in Figure 1 or 5 , could you add it?
Author Response
Replies to the comments of reviewer #2
Comment: Table I - Could you add the Oxygen concentration, or the measurement error, so as we can see, that the initial O content is below that value, which cannot be measured?
Answer:
- As mentioned by the reviewer, we analyzed oxygen concentration of the ingot by an oxygen analyzer and added following sentences.
“Oxygen concentration of the ingot was analyzed using an inert gas fusion infrared absorption method employing an oxygen analyzer (LECO, 736 series) with a graphite crucible. The oxygen concentration of the ingot after homogenization annealing was measured to 0.001wt%.” (line 76)
Comment: Line 174 - "To investigate the change in phase with respect to the sintering temperature" the word composition should be inserted.
Answer:
- As mentioned by the reviewer, we changed the sentence as follow.
“To investigate the change in phases and their composition with respect to the sintering temperature, the compacts sintered at 1500, 1600, and 1700°C were analyzed by XRD, and the results are shown in Fig. 6(a).” (line 203)
Comment: Table 2 contains data from the sintered compacts, which are discussed later in the text. Could you please the table after Figure 6? The table shows wt% of phases, please add this information, that these are wt% also to the text (Line 206-210)
Answer:
- As mentioned by the reviewer, we moved Table II behind Fig. 6.
- A mentioned by the reviewer, we added following sentences.
“Table II. The percentage of phases in the ingot and the compacts sintered at 1500, 1600, and 1700°C analyzed using the Rietveld method [wt%].” (line 224)
“The phase percentages of Table 2 were represented by wt%.” (line 237)
Comment: Line 181-182- (ICDD # xx-xx-xxx) - should be deleted, The lattice parameters are in Angstroms not nanometers
Answer:
- I wrote the ICDD number because another reviewer wants to present the code number.
- The unit of the lattice parameter was changed from nanometers to Angstroms.
“However, lattice parameters a, b, and c of α-Nb5Si3 are 6.56, 6.56, and 11.86 Å, respectively (ICDD # 98-001-6774), while those of β- Nb5Si3 are 10.02, 10.02, and 5.07 Å, respectively (ICDD # 98-064-5411).” (line 121)
Comment: You discussed at several places, that the Si phase oxidizes, why there are no maps for O content in Figure 1 or 5 , could you add it?
Answer:
- In the high temperature oxidation behavior of the Nb silicide based alloy, SiO2 is firstly formed on the silicide phase, and Si is diffused into the Nb matrix to form an oxidation resistant film on the whole surface. Therefore, oxide layer formed on the silicide phase could be observed after the oxidation heat treatment at high temperatures; however the Fig. 1 and 5 show the polished microstructure. Also, no difference occurred in the result of oxygen mapping through EDS.

Reviewer 3 Report
This article is interesting. The article’s topic is addressed in an appropriate, consistent and thorough way.
This paper reports on the synthesis of Nb-6Mo-20Si-3Cr (at%) in situ composite powders. The alloy powders were fabricated by a hydrogenation-dehydrogenation method, and both the ingot and powders were composed of two phases: a Nb solid solution phase and the α-Nb5Si3 phase. The phase composition, crystalline structure and microstructure of consolidated alloy powders at 1500, 1600 and 1700°C using spark plasma sintering, were studied in details.
Before publishing in Materials I think the authors should explain the following comments:
Line 96 - the XRD pattern was composed of two phases, the Nb solid solution phase and the α-Nb5Si3 silicide phase – please add the ref. codes (ICCD)
Lines 98-100 - Table II shows the percentages of the two phases in the ingot, which were determined from the XRD data using the Rietveld method – what was the fitting factor (Rwp)?
It will be useful to add the paper the XRD pattern of the hydrogenated alloy
Line 130 - What was the volume expansion of the alloy after the hydrogenation ?
Lines 181-182 – ICCD codes are missing
Author Response
Replies to the comments of reviewer #3
Comment: Line 96 - the XRD pattern was composed of two phases, the Nb solid solution phase and the α-Nb5Si3 silicide phase – please add the ref. codes (ICCD)
Answer:
- As mentioned by the reviewer, we added ICCD code of phases.
“The XRD spectrum was mainly composed of the Nb solid solution phase (ICDD # 98-064-5060) and the Nb5Si3 phases.” (line 207)
“However, lattice parameters a, b, and c of α-Nb5Si3 are 6.56, 6.56, and 11.86 Å, respectively (ICDD # 98-001-6774), while those of β-Nb5Si3 are 10.02, 10.02, and 5.07 Å, respectively (ICDD # 98-064-5411).” (line 210)
Comment: Lines 98-100 - Table II shows the percentages of the two phases in the ingot, which were determined from the XRD data using the Rietveld method – what was the fitting factor (Rwp)?
Answer:
- As mentioned by the reviewer, we added following sentence.
“The weighted profile R-factors (Rwp) in the Rietveld analysis were ranged from 15 to 19%.” (line 237)
Comment: It will be useful to add the paper the XRD pattern of the hydrogenated alloy.
Answer:
- As mentioned by the reviewer, we added XRD pattern of the hydrogenated ingot in Fig. 2 and following sentences.
“After annealing in the hydrogen atmosphere, the Nb solid solution phase was hydrogenated and its peak positions were changed; however, in the case of the Nb5Si3 phase, the positions of the peaks were maintained. After removal of hydrogen through dehydrogenation annealing, the XRD pattern was returned to the same as the initial ingot.” (line 117)
Comment: Line 130 - What was the volume expansion of the alloy after the hydrogenation ?
Answer:
- As mentioned by the reviewer, we added the following sentences.
“Based on the XRD results (Fig. 2), the lattice parameter of Nb phase in the ingot was 3.263 Å and it was increased to 3.372 Å after the hydrogen annealing. Thus, the volume of about 10 % was expanded by hydrogen absorption after the hydrogenation.” (line 149)
Comment: Lines 181-182 – ICDD codes are missing.
Answer:
- As mentioned by the reviewer, we added ICDD codes.
“However, lattice parameters a, b, and c of α- Nb5Si3 are 6.56, 6.56, and 11.86 Å, respectively (ICDD # 98-001-6774), while those of β-Nb5Si3 are 10.02, 10.02, and 5.07 Å, respectively (ICDD # 98-064-5411).” (line 210)

Round 2
Reviewer 1 Report
Acceptable for publication now.